# Non-compositional Expression Generation Based on Curriculum Learning and Continual Learning

**Jianing Zhou[1], Ziheng Zeng[1], Hongyu Gong[2], Suma Bhat[1]**
[1]University of Illinois Urbana-Champaign
[2]Facebook AI
[1]{zjn1746,zzeng13,spbhat2}@illinois.edu
[2]hygong@fb.com

## Abstract

Non-compositional expressions, by virtue of their non-compositionality, are a classic 'pain in the neck' for NLP systems. Different from the general language modeling and generation tasks that are primarily compositional, generating non-compositional expressions is more challenging for current neural models, including large pre-trained language models. The main reasons are 1) their non-compositionality, and 2) the limited data resources. Therefore, to make the best use of available data for modeling non-compositionality, we propose a dynamic curriculum learning framework, which learns training examples from easy ones to harder ones thus optimizing the learning step by step, but suffers from the forgetting problem. To alleviate the forgetting problem brought by the arrangement of training examples, we also apply a continual learning method into our curriculum learning framework. Our proposed method combined curriculum and continual learning, to gradually improve the model's performance on the task of non-compositional expression generation. Experiments on idiomatic expression generation and metaphor generation affirm the effectiveness of our proposed curriculum learning framework and the application of continual learning. Our codes are available at `https://github.com/zhjjn/CL2Gen.git`.

## 1 Introduction

Natural language has a common yet special class of constructions called non-compositional expressions that exhibit *semantic non-compositionality*, where the meaning of the expression cannot be inferred from that of its constituent words (e.g., metaphors and idioms) (Baldwin and Kim, 2010). They are commonly used for specific communicative intents (Moon et al., 1998; Baldwin and Kim, 2010) and are individually rare but collectively frequent, appearing frequently across genres (Moon et al., 1998; Haagsma et al., 2020). Most of the idioms indexed by the Oxford Dictionary have a frequency of less than 1 per million in the corpus of Contemporary American English (Rafatbakhsh and Ahmadi, 2019). They have been classically regarded as a "pain in the neck" to NLP systems (Sag et al., 2002) not only because of their non-compositionality, but also because of their contextual semantic ambiguity (used in non-compositional or compositional meaning depending on the context). Different NLP tasks related to non-compositional expressions have been studied, including sentiment analysis (Biddle et al., 2020), paraphrase generation (Zhou et al., 2021c), natural language inference (Chakrabarty et al., 2021a), metaphor detection (Su et al., 2020) and idiom usage recognition (Liu and Hwa, 2018). However, the generation of non-compositional expressions remains an important yet under-explored problem. Therefore, this paper focuses on the generation of non-compositional expressions.

As is shown in Table 1, non-compositional expression generation aims to generate the correct non-compositional expression given a sentence with original non-compositional expression masked. Its importance stems from that **1)** non-compositional expressions are an important part of everyday human language use, and **2)** their use imparts naturalness, fluency and stylistic enhancement. Therefore, the ability to generate non-compositional expressions renders machine generated language more natural and human-like whereas current SOTA pre-trained text generation models only pre-trained on normal compositional language tend to only generate compositional expressions (Zeng and Bhat, 2022). Besides, in our experiments, the simply fine-tuned model cannot correctly generate the idioms most of the time. Previously only a few of studies focus on metaphor generation (Yu and Wan, 2019; Chakrabarty et al., 2020; Stowe et al., 2021) whereas other types of non-compositional expressions remain under-

| Idiom |
|---|
| *Input Sentence* |
| It looks like the temperature is going to drop tonight , so be careful not to **[MASK]** . |
| *Output Sentence* |
| It looks like the temperature is going to drop tonight , so be careful not to **catch a cold** . |
| **Metaphor** |
| *Input Sentence* |
| The scream **[MASK]** the night . |
| *Output Sentence* |
| The scream **pierced** the night . |

Table 1: Examples of input and output in our tasks. Non-compositional expressions are highlighted in **bold red**

explored (e.g. idioms). The sparsity of literature and data resources presents challenges for the study of non-compositional expression generation.

To better utilize available data and alleviate the limitation on resources, curriculum learning (Bengio et al., 2009) aims to enable the models to begin training from easier examples proceeding to examples with an increasing level of difficulty. As such, curriculum learning consists of two core constituents: (1) Deciding the level of learning difficulty for each example, and (2) Scheduling the order of training examples based on that difficulty level. Curriculum learning has recently emerged as a promising direction for different fields including computer vision (Weinshall et al., 2018; Wang et al., 2019; Li et al., 2020) and natural language processing (Platanios et al., 2019; Liu et al., 2020; Zhou et al., 2021d; Zhang et al., 2021). However, despite the relative success on computer vision tasks, the application of curriculum learning for natural language processing is still limited to neural machine translation, which has rich data resources while other applications, such as non-compositional expression generation, with limited data remain under-explored.

To this end, we propose a novel curriculum learning framework for non-compositional expression generation to fill the research gap of generating non-compositional expressions including both metaphors and idioms. Our study is the first to focus on this task and utilizes curriculum learning to alleviate the problem caused by limited data resources. In our work, we use the *representation distance* and the *perplexity score* as the difficulty measurement and a dynamic scheduling method to order the examples. Specifically, we observe that curriculum learning orders the training examples according to difficulty level to create a gradual shift of distribution of domain difficulty, which will cause the well-known *catastrophic forgetting* prob-

lem (French, 1993) ignored in previous curriculum learning works. Therefore, we propose RE-GEM, a continual learning algorithm to alleviate the forgetting of learned knowledge in the early stage.

Overall, the main contributions are as follows:

- We conduct a first study on non-compositional expression generation including both metaphors and idioms.

- We propose a novel curriculum learning framework specifically designed for non-compositional expression generation that uses the distance between contextualized representations and word embeddings and perplexity score as a measure of difficulty level. It is dynamically updated with the training, based on which the training examples are scheduled.

- We point out for the first time the forgetting problem caused by the curriculum learning and propose a scheme—RE-GEM—to alleviate this problem.

- We evaluate our proposed framework on two tasks: idiomatic expression generation and metaphor generation. Experimental results on both tasks affirm the effectiveness of our framework. Detailed ablation studies and analysis are provided to support our claims.

## 2 Related Work

**Non-compositional Expression.** As an integral part of natural language, non-compositional expressions are classically regarded as a "pain in the neck" for NLP (Sag et al., 2002) due to their non-compositionality. Prior studies mainly focused on tasks related to non-compositional expressions, including identifying potentially idiomatic expressions (Salehi et al., 2014; Senaldi et al., 2016; Flor and Klebanov, 2018; Amin et al., 2021; Zeng and Bhat, 2021), disambiguating between their figurative/literal use (Peng and Feldman, 2015; Köper and im Walde, 2016; Liu and Hwa, 2017, 2018), detecting metaphors (Gao et al., 2018; Mao et al., 2019; Su et al., 2020; Gong et al., 2020), generating metaphors (Yu and Wan, 2019; Stowe et al., 2020; Chakrabarty et al., 2020; Stowe et al., 2021) and paraphrasing between non-compositional expressions and their literal counterparts (Liu and Hwa, 2016; Agrawal et al., 2018; Shirin and Raseek, 2018; Zhou et al., 2021a,b). However, owing to their non-compositionality and limitations on available and related data resources (Stowe et al., 2020,

2021), the task of generating non-compositional expressions including both idioms and metaphors remains challenging and under-explored. Our study first focuses on alleviating the limitation on resource availability concerning the generation of both idioms and metaphors.

**Curriculum Learning.** First proposed by (Bengio et al., 2009), curriculum learning enables machine learning model training to gradually proceed from easy examples to harder ones according to a measure of difficulty level for each example, thereby permitting a better utilization of available data resources. With growing research interests received, curriculum learning has been applied to different fields including computer vision (Weinshall et al., 2018; Wang et al., 2019; Li et al., 2020) and natural language processing. Despite its benefits observed in computer vision tasks, including image classification (Weinshall et al., 2018), human attribute analysis (Wang et al., 2019) and visual question answering (Li et al., 2020), it has seen limited applicability in NLP mainly to NMT (Platanios et al., 2019; Liu et al., 2020; Zhou et al., 2021d). As a result, curriculum learning methods, including difficulty measurement and scheduling strategies, are mainly designed for the NMT task, which is largely different from the task of processing non-compositionality (non-compositional expression generation). To this end, we propose our curriculum learning method specifically designed for non-compositional expression generation.

**Continual Learning** Continual learning enables models to learn new knowledge and preserve knowledge acquired previously from a data stream with a continuously changing distribution. However, due to the well-known problem of *catastrophic forgetting* (French, 1993), continual learning is still challenging for current neural models. The same forgetting problem could also appear in curriculum learning because curriculum learning rearranges the examples according to their difficulty levels, which will naturally create a training data stream with continuously changing distribution on difficulty domain and thus cause the problem of forgetting. Although the catastrophic forgetting problem has been explored in both computer vision (Rebuffi et al., 2017; Kirkpatrick et al., 2017; Zenke et al., 2017; Aljundi et al., 2019) and natural language processing (Xu et al., 2018; Liu et al., 2019; Sun et al., 2019; Chen et al., 2015; Shu et al., 2016; Thompson et al., 2019), there are no avail-

able studies on curriculum learning mentioning this forgetting problem. Our work is the first attempt to point out the issue of forgetting in curriculum learning and study mechanisms to alleviate it.

## 3 Framework

In this section, we briefly introduce our proposed curriculum learning method for non-compositional expression generation.

Curriculum learning for efficiently leveraging available data resources consists of two main parts: a measure of difficulty of training instances, and an arrangement of the training examples using this measure. Accordingly, for non-compositional expression generation, we propose a data arrangement method for dynamically arranging the training examples according to a newly studied difficulty metric. In addition, due to the current large pre-trained language models' insufficiency in processing non-compositional expressions (Dankers et al., 2022), non-compositional expressions that are difficulty for LMs to understand would have a high perplexity score and the representations between non-compositional expressions and their constituent words would be large. Therefore, we use a combination of the representation distance and perplexity score as a measure of examples' difficulty.

Moreover, in our experiments, we observe that following the curriculum learning principle of arranging the training examples based on their difficulty levels, the problem of forgetting arises due to the gradual shift of distribution in domain difficulty. Therefore, to alleviate this forgetting problem, we propose a simple yet effective continual learning method. Figure 1 demonstrates the workflow of our proposed curriculum learning framework and its details as follows.

### 3.1 Difficulty Metrics

In this section, we define the difficulty metric used by our framework. Previous works on curriculum learning mainly focus on compositional languages. Therefore, difficulty metrics, including sentence length, word rarity, embedding norm and etc, proposed in prior works cannot reflect the difficulty levels of non-compositional expressions in the sentences. As mentioned in Section 1, due to the non-compositionality, the meaning of the non-compositional expressions is different from the meaning of the constituent words. Therefore, the distance between the ideal representation of the

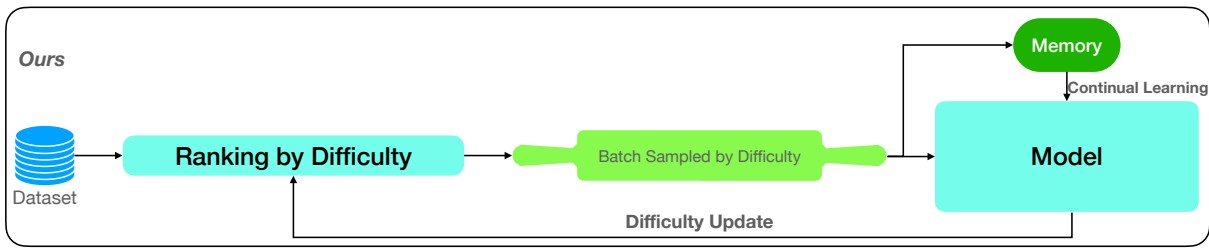

Figure 1: The overview of our framework and the comparison between our framework and other CL methods.

non-compositional expressions and the representations of the constituent words would be large. Utilizing this property, we first propose to use the distance between the contextualized representations of the non-compositional expressions and the original word embeddings of the constituent words to reflect the difficulty. A larger distance represents the model has already learned the representation of the expression instead of using the constituent words' embeddings, which means this non-compositional expression is easy for the model. On the contrary, a smaller distance means the target non-compositional expression is difficulty for the model. Therefore, the difficulty metric based on representation distance is calculated as follows:

$$d^r(Y) = \frac{1}{\text{DIST}(Y;\theta)} = \frac{1}{\| l(Y) - Emb(Y) \|}$$

where $l(\cdot)$ is the final layer of the model and $Emb(\cdot)$ is the embedding layer of the model.

Besides, as mentioned in Section 1, due to the rareness of non-compositional expressions in large scale corpora relative to compositional expressions, large pre-trained models seldom see the use of non-compositional expressions during pre-training, which results in their inability to accurately capture the semantics of these expressions. Therefore, we assess the difficulty of training examples based on the models' familiarity with the non-compositional expressions. Toward this, we propose to utilize the *perplexity score* as a measure of the difficulty of each training example. This stems from the idea that in language modeling, perplexity is used as a quality measure for language models, which is indicative of its ability to predict the next word in a sequence of words. Perplexity score is built with n-grams that are extracted from text corpora: a lower perplexity score of an n-gram $X$, i.e.,

$$\text{PPL}(X) = e^{-\frac{1}{t} \sum_i^t \log p_\theta(x_i|x_{<i})},$$

means the language model assigns a higher probability to generating $X$. Therefore, a lower perplexity on a non-compositional expression is indicative

that the language model is more familiar with this expression, i.e., this expression is easier for the language model and thus is more likely to generate it. The difficulty metric based on *perplexity score* is calculated as follows:

$$d^p(Y) = \text{PPL}(Y;\theta) = e^{-\frac{1}{t} \sum_i^t \log p_\theta(y_i|y_{<i})}$$

where $Y$ is the target sentence in a training example and $\theta$ is the trainable parameters.

### 3.2 Scheduling Strategy

Having ascertained the difficulty level for each training example, the traditional curriculum learning scheme re-arranges the training examples using the difficulty level and fixes the order of the examples for the subsequent training process. However, after training the model on some training examples, it is expected that the perceived difficulty level of each training example will change. Therefore, it is unreasonable to use the same order of training examples for the entire training process. To address this issue, a competence score is proposed to dynamically reflect the model's ability. However, the competence score used by previous works, such as a number increased with time steps (Platanios et al., 2019), is actually not comparable to the difficulty score of training examples because the two measure unrelated aspects. To better reflect the dynamic difficulty levels, we propose a dynamic scheduling method that arranges the training examples.

After each training epoch, the difficulty score $d$ for each training example is updated using the model trained in the most recent epoch:

$$d_n(Y) = d^r(Y) + d^p(Y)$$

where $d_n(Y)$ is the difficulty score for training example $(X;Y)$ after the model has been fine-tuned for $n$ epochs and $\theta_n$ refers to the trainable parameters of the model that has been fine-tuned for $n$ epochs. $X$ represents the input sentence with the target non-compositional expression masked in the training example and $Y$ is the target sentence. After the difficulty scores for all the training examples

**Algorithm 1: PPLCL**

**Input:** Dataset $\mathbb{P}$, Model $\mathbf{M}$ and number of epochs $N$
**Output:** Fine-tuned Model $\mathbf{M}^*$

1  $\mathbf{D}_0 = \mathcal{D}(\mathbb{P}, \mathbf{M})$ ;
2  Sort $\mathbb{P}$ based on each difficulty level in $\mathbf{D}_0$, resulting in a re-arranged $\mathbb{P}_0$ ;
3  **for** $n = 1; n \leq N$ **do**
4      $\mathbf{M}_{\theta_n} \Leftarrow \text{TRAIN}(\mathbb{P}_{n-1})$;
5      $\mathbf{D}_n = \varnothing, \mathbb{P}_n^* = \varnothing$ ;
6      **for** $(X; Y) \in \mathbb{P}$ **do**
7         $d_n(Y) = d^r(Y) + d^p(Y)$ ;
8         **if** $d_n(Y) \neq d_{n-1}(Y)$ **then**
9            $\mathbf{D}_n \Leftarrow \mathbf{D}_n \bigcup \{d_n(Y)\}$ ;
10           $\mathbb{P}_n^* \Leftarrow \mathbb{P}_n^* \bigcup (X; Y)$ ;
11        **else**
12           continue ;
13        **end**
14     **end**
15     Sort $\mathbb{P}_n^*$ based on $\mathbf{D}_n$, resulting in $\mathbb{P}_n$ ;
16 **end**
17 **return** $\mathbf{M}^* = \mathbf{M}_{\theta_n}$ ;

---

**Algorithm 2: TRAIN**

**Input:** Train Dataset $\mathbb{P}$, Model $\mathbf{M}_\theta$
**Output:** Fine-tuned Model $\mathbf{M}^*$

1  $\mathcal{M} \leftarrow \{\}$ ;
2  **for** $t = 1; t \leq T$ **do**
3      **for** $(\mathbf{X}, \mathbf{Y}) \in \mathbb{P}_t$ **do**
4         $\mathcal{M}_{ref} = (\mathbf{X}_{ref}, \mathbf{Y}_{ref}) \sim \mathcal{M}$ ;
5         $g \leftarrow \nabla_\theta l(\mathbf{M}_\theta(\mathbf{X}), \mathbf{Y})$ ;
6         $g_{ref} \leftarrow \nabla_\theta l(\mathbf{M}_\theta(\mathbf{X}_{ref}), \mathbf{Y}_{ref})$ ;
7         **if** $g^\top g_{ref} \geq 0$ **then**
8            $\theta \leftarrow \theta - \alpha g$ ;
9         **else**
10           $\tilde{g} \leftarrow g_{ref} - \frac{g_{ref}^\top g}{g^\top g} g; \theta \leftarrow \theta - \alpha(g + \tilde{g})$;
11        **end**
12     **end**
13     $s \leftarrow \frac{|\mathcal{M}|}{T}$ ;
14     **for** $i = 1; i \leq s$ **do**
15        $(\mathbf{X}, \mathbf{Y}) \sim \mathbb{P}_t; \mathcal{M} \leftarrow (\mathbf{X}, \mathbf{Y})$
16     **end**
17 **end**
18 **return** $\mathbf{M}^* = \mathbf{M}_\theta$ ;

---

have been updated, the training examples will be re-arranged according to the new difficulty scores.

### 3.3 Continual Learning

Essentially, after the order of the training examples is re-arranged based on difficulty level, the curriculum learning scheme creates a gradual shift of the distribution in the domain difficulty, which will cause the problem of *catastrophic forgetting*, currently ignored by previous studies of curriculum learning (Bengio et al., 2009; Weinshall et al., 2018; Wang et al., 2019; Li et al., 2020; Platanios et al., 2019; Liu et al., 2020; Zhou et al., 2021d).

During training, the model will first learn the examples with a lower difficulty level and then learn

those with a higher difficulty in each epoch. In this process, some knowledge about the examples with a lower difficulty level will be forgotten.

To alleviate the forgetting problem, we propose RE-GEM, a modified version of GEM (Lopez-Paz and Ranzato, 2017), which fits in our framework better compared with the traditional continual learning methods for the following reason.

The traditional continual learning methods like GEM aim to alleviate the forgetting problem and thus sacrifice the learning ability on new data and part of the overall performance. This is done with the use of an episodic memory, $\mathcal{M}_k$, containing randomly sampled training examples from the data $\mathbb{P}_k$ for the time step $k$. When minimizing the loss on the current time step $t$, GEM treats the losses on $\mathcal{M}_k$ of step $k < t$ as constraints by preventing their increase. An improved version of GEM (Chaudhry et al., 2018) was proposed to only treat the loss on a subset $\mathcal{M}_{ref}$ of all the episodic memories for step $k < t$ as a constraint instead of computing multiple losses. To guarantee the loss reduction on this episodic memory subset, $\mathcal{M}_{ref}$, their implementation first computes the loss gradient vector $g$ on the current step and then computes the loss gradient vector $g_{ref}$ on the subset $\mathcal{M}_{ref}$. Whenever the angle between $g$ and $g_{ref}$ is greater than 90°, the gradient $g$ will be projected to $\hat{g}$ as:

$$\text{minimize}_{\hat{g}} \frac{1}{2} \parallel g - \hat{g} \parallel_2 \quad \text{s.t.} \quad \hat{g}^\top g_{ref} \geq 0$$

and the parameters will be updated based on $\hat{g}$, with the intent of avoiding the forgetting of previous data and learning from new data.

Instead, our main focus is to learn from new data and then to alleviate the forgetting of previous data through the use of RE-GEM. Therefore, when the angle between $g$ and $g_{ref}$ is greater than 90°, we first project $g_{ref}$ to $\tilde{g}$ as follows:

$$\text{minimize}_{\tilde{g}} \frac{1}{2} \parallel g_{ref} - \tilde{g} \parallel_2 \quad \text{s.t.} \quad \tilde{g}^\top g \geq 0 \quad (1)$$

Then the parameters will be updated based on both $g$ and $\tilde{g}$ where $g$ guarantees the successful learning of current new data and $\tilde{g}$ tries best to alleviate the forgetting of previous data. We leave exploring other forms of gradient updates to future work.

The constraint optimization problem in Eq.1 can be solved via the rule proposed in (Chaudhry et al., 2018) as follows:

$$\tilde{g} = g_{ref} - \frac{g_{ref}^\top g}{g^\top g} g.$$

| Methods | 1 Epoch | | | | | 5 Epochs | | | | |
|---|---|---|---|---|---|---|---|---|---|---|
| | Acc | BLEU | phrase-BLEU | Rouge | phrase-Rouge | Acc | BLEU | phrase-BLEU | Rouge | phrase-Rouge |
| Vanilla | 18 | 56.65 | 37.74 | 69.94 | 40.69 | 25 | 61.87 | 43.82 | 73.27 | 48.81 |
| Competence + SL | 16 | 53.82 | 35.02 | 67.90 | 37.81 | 24 | 61.11 | 43.42 | 72.62 | 47.77 |
| Competence + WR | 16 | 54.01 | 35.87 | 68.09 | 38.11 | 24 | 61.16 | 44.18 | 72.91 | 47.56 |
| Norm-based | 16 | 54.12 | 35.43 | 67.96 | 38.02 | 24 | 61.34 | 44.22 | 73.06 | 48.04 |
| Fixed SGCL | 19 | 56.25 | 37.48 | 69.93 | 41.83 | 24 | 60.11 | 43.24 | 72.35 | 47.63 |
| Dynamic SGCL | 17 | 54.56 | 37.23 | 68.72 | 38.92 | 22 | 59.37 | 41.07 | 72.08 | 46.09 |
| Ours | **23** | **60.43** | **43.72** | **72.31** | **45.77** | **28** | **64.36** | **48.02** | **74.94** | **51.82** |
| *p-value* | 0.02 | 0.01 | 0.008 | 0.007 | 0.006 | 0.01 | 0.005 | 0.005 | 0.01 | 0.006 |

Table 2: Performance of different methods on MAGPIE dataset. **Competence** represents using competence score for scheduling. **SL** refers to using sentence length as difficulty score. **WR** refers to using word rarity as difficulty score. Best performance is labeled in bold. Models trained for 5 epochs are converged. *p-value* refers to the results of significance test based on our method and second best method (Fixed SGCL).

## 4 Experiments

### 4.1 Datasets

We use two datasets focusing on two kinds of non-compositional expressions—MAGPIE (Haagsma et al., 2020) for idiom and MERMAID (Chakrabarty et al., 2021b) for metaphor. For the instances from MAGPIE, we mask the target idiom in each example. The position of the target idiom is provided in the dataset. The official training-development-testing splits are used. For MERMAID, the masked sentences have been provided and we use the available data splits.

### 4.2 Baselines

We tested six baseline models compare them with our proposed curriculum learning framework. The Vanilla model that does not use any CL methods, Competence-based CL (Platanios et al., 2019), Norm-based CL (Liu et al., 2020) and SGCL (Zhou et al., 2021d) are used as baselines. Due to space limitation, their description and experimental settings are provided in the Appendix.

### 4.3 Experimental Settings

For our framework, we utilize BART-base as our backbone model. For the task of idiomatic expression generation, the model is trained with batch size of 8 for 5 epochs. For metaphor generation, the model is trained with batch size of 16 for 10 epochs. Adam optimizer is used and the learning rate is set to $5 \times 10^{-5}$. All the other parameters are set to their default. All of our experiments are performed 5 times and the mean of the results are reported. Beam search is used when decoding.

### 4.4 Evaluation Metrics

**Automatic Evaluation** Considering our focus non-compositional expression generation, we use the widely used text generation evaluation metrics

*ROUGE* (Lin, 2004) and *BLEU* (Papineni et al., 2002) for evaluation. To automatically evaluate how well the idiomatic expressions are generated, we extract the newly generated part in the output sentences and then compare it with the target non-compositional expressions in the references via phrase-level BLEU and ROUGE scores following (Zhou et al., 2021c). We also evaluate with a stricter metric of phrase-level *Accuracy*, in which the generated non-compositional expression is considered to be correct if and only if every word strictly matches the target expression. For metaphoric expression generation task, corpus-level BLEU score and ROUGE score are used for evaluation following (Chakrabarty et al., 2021b). Due to the fact that the target metaphoric expressions only contain one word and is measured by ROUGE-1 score, we did not use the phrase-level scores described above.

**Human Evaluation** We used 100 instances from test sets for both tasks, and collected the outputs from the 3 best methods ranked by automatic evaluation. For each output sentence, two native English speakers, who were blind to the systems being compared, were asked to rate the output sentences. For idiom generation, we propose the following criteria: (1) Meaning ("Are the output and the reference meaning the same thing?") (2) Fitness ("Does the generated idiom make sense in the context?") (3) Fluency ("How fluent, grammatical, well formed and easy to understand are the generated utterances?") (4) Overall ("What is the overall quality of the generated utterances?"). For metaphor generation, criteria described in (Chakrabarty et al., 2021b) is used. More details are in the Appendix.

## 5 Results

As shown in Table 2, for the idiomatic expression generation task, our proposed framework achieves the best performance with respect to all the evalu-

| Methods | 1 Epoch | | | | | | 10 Epochs | | | | | |
|---|---|---|---|---|---|---|---|---|---|---|---|---|
| | BLEU | BLEU-2 | BLEU-4 | Rouge-1 | Rouge-2 | Rouge-L | BLEU | BLEU-2 | BLEU-4 | Rouge-1 | Rouge-2 | Rouge-L |
| Vanilla | 50.53 | 64.54 | 40.51 | 78.13 | 63.51 | 78.12 | 53.96 | 67.21 | 44.80 | 79.42 | 66.06 | 79.41 |
| Competence + SL | 49.55 | 63.88 | 39.04 | 77.80 | 62.82 | 77.78 | 50.21 | 64.21 | 39.69 | 78.02 | 63.17 | 78.01 |
| Competence + WR | 49.19 | 63.46 | 38.59 | 77.61 | 62.55 | 77.59 | 53.76 | 65.78 | 42.90 | 78.81 | 65.09 | 78.80 |
| Norm-based | 49.02 | 63.23 | 38.44 | 77.42 | 62.35 | 77.43 | 53.54 | 65.55 | 42.68 | 78.60 | 64.88 | 78.65 |
| Fixed SGCL | 50.62 | 64.57 | 40.60 | 78.42 | 63.55 | 78.43 | 54.14 | 67.55 | 44.88 | 79.60 | 66.88 | 79.65 |
| Dynamic SGCL | 49.3 | 63.54 | 39.42 | 77.63 | 63.11 | 78.03 | 53.65 | 66.27 | 43.66 | 79.13 | 65.83 | 78.95 |
| Ours | **51.61** | **65.46** | **42.12** | **78.58** | **64.42** | **78.57** | **57.94** | **69.88** | **46.77** | **81.54** | **68.49** | **81.54** |
| *p-value* | 0.005 | 0.006 | 0.002 | 0.003 | 0.004 | 0.007 | 0.006 | 0.007 | 0.003 | 0.006 | 0.007 | 0.008 |

Table 3: Performance of different methods on MERMAID dataset. Best performance is labeled in bold.

| Methods | 1 Epoch | | | | | 5 Epochs | | | | |
|---|---|---|---|---|---|---|---|---|---|---|
| | Acc | BLEU | phrase-BLEU | Rouge | phrase-Rouge | Acc | BLEU | phrase-BLEU | Rouge | phrase-Rouge |
| Vanilla | 18 | 56.65 | 37.74 | 69.94 | 40.69 | 25 | 61.87 | 43.82 | 73.27 | 48.81 |
| Diff + Fixed | 20 | 57.36 | 38.10 | 70.22 | 42.52 | 25 | 62.05 | 45.20 | 73.40 | 49.40 |
| Diff + Competence | 20 | 58.33 | 39.12 | 70.43 | 42.73 | 25 | 62.44 | 46.21 | 73.82 | 49.73 |
| SL + Competence | 16 | 53.82 | 35.02 | 67.90 | 37.81 | 24 | 61.11 | 43.42 | 72.62 | 47.77 |
| WR + Competence | 16 | 54.01 | 35.87 | 68.09 | 38.11 | 24 | 61.16 | 44.18 | 72.91 | 47.56 |
| Norm + Competence | 16 | 54.12 | 35.43 | 67.96 | 38.02 | 24 | 61.34 | 44.22 | 73.06 | 48.04 |
| Diff + Dynamic | 20 | 59.06 | 40.08 | 71.31 | 43.42 | 26 | 63.20 | 46.84 | 74.57 | 50.17 |
| Diff + Dynamic + Re-GEM | **23** | **60.43** | **43.72** | **72.31** | **45.77** | **28** | **64.36** | **48.02** | **74.94** | **51.82** |
| Diff + Dynamic + ER | 18 | 53.53 | 34.16 | 66.93 | 37.55 | 18 | 53.63 | 34.89 | 67.02 | 37.93 |
| Diff + Dynamic + MIR | 21 | 59.06 | 40.48 | 71.58 | 43.71 | 25 | 63.32 | 45.53 | 74.39 | 49.86 |
| Diff + Dynamic + AGEM | 21 | 58.66 | 40.61 | 71.28 | 43.01 | 25 | 62.36 | 45.00 | 73.44 | 49.27 |

Table 4: Ablation study on MAGPIE dataset. **Diff** refers to our difficulty metric. **Fixed** means the training examples are sorted only once before training and fixed during training. **Dynamic** refers to our dynamic scheduling strategy.

ation metrics. Compared with the performance of the vanilla model, our framework outperforms it by 5 on accuracy, 3.78 on BLEU, 5.98 on phrase-level BLEU, 2.37 on Rouge and 5.08 on phrase-level ROUGE score after only 1 training epoch. After 5 training epochs, the improvements after convergence are 3 on accuracy, 2.49 on BLEU, 4.2 on phrase-level BLEU, 1.67 on ROUGE and 3.01 on phrase-level ROUGE.

For other baseline models, it is obvious that most of them are not competitive compared with the vanilla model. Some of the baseline methods even degrade the performance of the vanilla model.

Table 3 presents the results on the task of metaphor generation task. As in the case of idiomatic expression generation, in this task our proposed framework achieves the best performance considering all the evaluation metrics. Compared with the performance of vanilla model, our framework outperforms it by 1.08 on BLEU, 0.92 on BLEU-2, 1.71 on BLEU-4, 0.45 on ROUGE-1, 0.91 on ROUGE-2 and 0.45 on ROUGE-L after only 1 training epoch. After 10 training epochs, the improvements increase to 3.98 on BLEU, 2.67 on BLEU-2, 1.97 on BLEU-4, 2.12 on ROUGE-1, 2.43 on ROUGE-2 and 2.13 on ROUGE-L. Table 5 presents the results of human evaluation. It is shown that our method still outperforms other baselines and vanilla model by large margin on both idiom and metaphor generation task.

It should be noted that for metaphor generation task, all the baseline curriculum learning methods' influence on the performance is similar to that in the idiomatic expression generation task. That is, most of the baseline methods do are not competitive compared with the vanilla model, whereas our proposed curriculum learning framework shows an obvious improvement over the vanilla model.

Based on the performance on both tasks, we see that the baseline curriculum learning methods cannot effectively improve the performance of the vanilla model (and may even negatively influence the performance). However, our proposed curriculum learning framework outperforms all the baseline models by reasonably large margins.

## 6 Analysis

Here we provide some ablation studies based on idiomatic expression generation to analyze the contribution of different modules used in our framework. **Difficulty measurement.** As shown in Table 8, using our difficulty metric can boost the performance of the vanilla model, which verifies the effectiveness of our proposed measurement of difficulty. Compared with the vanilla model, using our difficulty metric alone can improve the performance by 2 on accuracy, 0.71 on BLEU and 0.28 on ROUGE even without the scheduling method. In addition, compared with the other difficulty measurement methods used in previous studies (e.g. sentence length, word rarity and norm), ours shows a larger performance improvement (rows 3-6 in Table 8).

| Model | Idiom | | | | Metaphor | | | |
|---|---|---|---|---|---|---|---|---|
| | Meaning | Fitness | Fluency | Overall | Fluency | Meaning | Creativity | Metaphoricity |
| Vanilla | 1.88 | 1.58 | 3.97 | 2.48 | 3.51 | 3.52 | 3.3 | 3.08 |
| Fixed SGCL | 1.86 | 1.68 | 3.95 | 2.50 | 3.29 | 2.92 | 3.51 | 3.46 |
| Ours | **2.98** | **3.69** | **4.16** | **3.61** | **4.36** | **3.84** | **3.43** | **3.85** |

Table 5: Human evaluation results. The best performance is in bold.

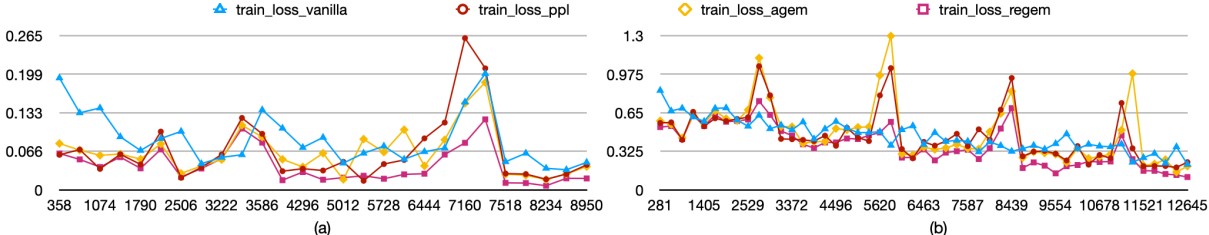

Figure 2: X-axis represents the training steps and Y-axis represents the training loss. (a) and (b) represents the results on MAGPIE dataset and MERMAID dataset respectively. 'vanilla' refers to the training loss of vanilla model. 'ppl', 'agem' and 'regem' refer to the training loss of the model using our CL methods without continual learning, with AGEM and our RE-GEM.

Using our difficulty metric can thus outperform the best among sentence length, word rarity and norm by 4 on accuracy, 4.21 on BLEU and 2.34 on ROUGE, demonstrating its superiority for non-compositional expression generation.

**Dynamic scheduling strategy.** The effectiveness of our dynamic scheduling strategy can be verified by comparing rows 2 and 7 in Table 8. We see that using our difficulty metric, the performance improves by 1.7 BLEU and 1.09 ROUGE points via dynamic scheduling compared with the performance when fixed scheduling is used, which confirms the effectiveness of our proposed dynamic scheduling method for curriculum learning.

**Continual learning scheme.** As stated in Section 3.3, curriculum learning will cause the forgetting problem, which has been ignored by previous studies. As shown in Figure 2, compared with the training loss of the vanilla model, the training loss of the model using only curriculum learning without continual learning shows sudden peaks at the beginning of each epoch. This shows that the model tends to forget knowledge learned from earlier (easier) examples in each epoch after curriculum learning is applied. Additionally, when the model moves from easier examples to harder examples, the knowledge learned from easier examples is beneficial for learning harder ones resulting in a non-increase of the training loss when the model learns harder examples during each epoch.

To alleviate the forgetting problem, we proposed a continual learning algorithm called RE-GEM to help with the learning. As shown in rows 7 and 8 in Table 8, the application of RE-GEM successfully improved the performance by 3 on accuracy, 1.37 on BLEU, 3.64 on phrase-level BLEU, 1 on ROUGE and 2.35 on phrase-level ROUGE scores when only perplexity score and dynamic scheduling are used. Besides, rows 8-11 show the advantage of our proposed RE-GEM over other continual learning methods, including ER (Robins, 1995), MIR (Aljundi et al., 2019) and AGEM (Chaudhry et al., 2018) when applied to curriculum learning. The performance of RE-GEM is better than the best among ER, MIR and AGEM by 3 on accuracy, 3.24 on phrase-level BLEU and 2.06 on phrase-level ROUGE, a trend persisting in Figure 2. When the training loss using curriculum learning with continual learning suddenly spikes, the peak of the one using RE-GEM is the lowest. This suggests our RE-GEM is more effective in alleviating the forgetting problem while maintaining the performance in curriculum learning, especially compared with the traditional continual learning methods.

## 7 Conclusion and Future Work

In this paper, we first utilize curriculum learning to better utilize available data for non-compositional expression generation. We propose a novel curriculum learning framework by utilizing representation distance and perplexity score as a measure of difficulty level and a dynamic scheduling method to better leverage the available training data. Furthermore, for the first time we study a continual learning algorithm to alleviate the forgetting problem resulting from curriculum learning. Experiments on

two non-compositional expression generation tasks of idiomatic expression generation and metaphor generation show that the proposed curriculum learning framework can effectively boost the performance of non-compositional expression generation outperforming previously studied curriculum learning methods. Future works should explore other difficulty metrics, more effective scheduling methods and continual learning schemes to further alleviate the forgetting problem, and study them for other text generation problems.

## 8 Limitations

As stated previously, our proposed framework utilizes perplexity score as a measure of difficulty, which is based on that non-compositional expressions are low resource languages compared with compositional expressions. Therefore, current large pre-trained language models will assign low probabilities to non-compositional expressions because of their unfamiliarity to non-compositional expressions. However, when it comes to compositional expressions, perplexity score cannot be used for measuring difficulty level, which limits our framework to only non-compositional expression generation. Besides, our scheduling strategy only re-schedules training examples after each training epoch instead of each batch, which also limits the flexibility of scheduling training examples. Therefore, the order of training examples in each training epoch will still be fixed. More flexible and dynamic scheduling strategies should be explored.

Another limitation lies in the gradient update in our RE-GEM. For gradient computed based on current data and the gradient computed based on data in the memory, we use the same learning rate to update them. This could be improved by setting different learning rates for gradients computed based on different data.

## Acknowledgements

We would like to acknowledge the assistance of Kellen Tan Cheng in performing the manual evaluation. This research was supported in part by the National Science Foundation under Grant No. IIS 2230817 and by a U.S. National Science Foundation and Institute of Education Sciences grant (2229612).

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

## A  Baseline Models

Details about the baseline models:

- **Vanilla**: The vanilla model directly use the pre-trained BART model for fine-tuning. For each training batch and epoch, random sampling is used to select training examples. No curriculum learning methods are applied.

- **Competence-based CL**: (Platanios et al., 2019) proposed to select training examples based on a competence score. Training examples with a difficulty score lower than current competence score will be selected as candidates for training. In their study, they proposed two measures of difficulty score: sentence length and word rarity.

- **Norm-based CL**: (Liu et al., 2020) proposed to use norm of word embeddings obtained from neural networks as a measure of both difficulty score and competence score. The selection of training examples is similar with the procedure described above in competence-based CL.

- **SGCL**: (Zhou et al., 2021d) proposed to utilize sentence-level BLEU score as a measure of the learning difficulty level. For the arrangement of training examples, the original training set will be divided into several mutual exclusive subsets according to difficulty level. Then they proposed two ways of scheduling: fixed scheduling and dynamic scheduling. We use both scheduling methods as our baseline models. More details about this baseline are described in (Zhou et al., 2021d).

## B  Implementation

Our experiments and implementation are based on the Transformers library and PyTorch.

## C  Experimental Details

All our experiments are conducted with 2 NVIDIA V100 GPUs.

## D  Human Evaluation

Here we provide more details of the human evaluation.

### D.1  Idiom Generation

Given reference sentences, annotators are expected to evaluate the quality of the generated sentences from three aspects:

1. Meaning: Are the output and the reference meaning the same thing? If yes, the score should be 3. If they are similar but not exactly the same, the score should be 2. If they are not similar, the score should be 1.

2. Fitness: Does the generated idiom make sense in the context? If the generated idiom always makes sense, the score should be 4. If the generated idiom makes sense under some circumstances, the score should be 3. If the generated idiom only makes sense under extreme circumstances, the score should be 2. If the generated idiom is invalid, the score should be 1.

3. Fluency: check whether the transferred sentence is fluent and readable on a scale of 1 to 5, ranging from "highly non-fluent" to "very fluent". This should include the tense (present or past), number(singular or plural) and pronoun(himself, herself, someone, his, her etc.)

### D.2  Metaphor Generation

Given input sentences and the reference sentences, annotators are expected to evaluate the quality of the generated sentences from four aspects. A scale of 1-5 where 1 denotes the worst and 5 be the best is used:

1. Fluency: How fluent, grammatical, well formed and easy to understand are the generated utterances?

2. Meaning: Are the input and the output referring or meaning the same thing?

3. Creativity: How creative are the generated utterances?

4. Metaphoricity: How metaphoric are the generated utterances

### D.3  Number of Parameters

Considering that our proposed curriculum learning and continual learning do not introduce more parameters, the number of parameters is identical to the number of parameters in the underlying language model: 140M for BART(base).

| Methods | 1 Epoch | | | | | 5 Epochs | | | | |
|---|---|---|---|---|---|---|---|---|---|---|
| | Acc | BLEU | phrase-BLEU | Rouge | phrase-Rouge | Acc | BLEU | phrase-BLEU | Rouge | phrase-Rouge |
| **Vanilla-BART-large** | 20 | 58.43 | 40.24 | 71.88 | 42.77 | 27 | 62.73 | 45.02 | 74.98 | 50.33 |
| **Diff + Dynamic -BART-large** | 22 | 60.47 | 42.83 | 74.02 | 45.71 | 28 | 64.55 | 47.83 | 77.02 | 53.11 |
| **Diff + Dynamic + Re-GEM -BART-large** | **25** | **61.73** | **44.24** | **75.12** | **46.83** | **29** | **66.02** | **49.13** | **78.94** | **55.03** |
| **Vanilla-BART-base** | 18 | 56.65 | 37.74 | 69.94 | 40.69 | 25 | 61.87 | 43.82 | 73.27 | 48.81 |
| **Diff + Dynamic -BART-base** | 20 | 59.06 | 40.08 | 71.31 | 43.42 | 26 | 63.20 | 46.84 | 74.57 | 50.17 |
| **Diff + Dynamic + Re-GEM -BART-base** | **23** | **60.43** | **43.72** | **72.31** | **45.77** | **28** | **64.36** | **48.02** | **74.94** | **51.82** |
| **Vanilla-T5-large** | 18 | 56.21 | 37.13 | 68.74 | 39.23 | 24 | 60.12 | 42.98 | 72.74 | 48.02 |
| **Diff + Dynamic -T5-large** | 20 | 57.94 | 39.85 | 70.03 | 41.73 | 25 | 61.03 | 45.02 | 74.38 | 50.12 |
| **Diff + Dynamic + Re-GEM -T5-large** | **22** | **60.74** | **42.23** | **71.94** | **43.72** | **27** | **62.91** | **47.03** | **76.55** | **51.82** |
| **Vanilla-T5-base** | 15 | 53.14 | 34.02 | 65.52 | 36.01 | 20 | 56.98 | 39.76 | 69.31 | 44.88 |
| **Diff + Dynamic -T5-base** | 17 | 54.26 | 36.13 | 66.74 | 37.82 | 21 | 58.83 | 41.77 | 70.69 | 46.72 |
| **Diff + Dynamic + Re-GEM** | **20** | **57.02** | **39.35** | **67.83** | **39.07** | **25** | **61.44** | **44.06** | **72.87** | **49.36** |

Table 6: Results based on different backbone model on MAGPIE dataset.

| Methods | 1 Epoch | | | | | | 10 Epochs | | | | | |
|---|---|---|---|---|---|---|---|---|---|---|---|---|
| | BLEU | BLEU-2 | BLEU-4 | Rouge-1 | Rouge-2 | Rouge-L | BLEU | BLEU-2 | BLEU-4 | Rouge-1 | Rouge-2 | Rouge-L |
| **Vanilla BART-large** | 50.53 | 64.54 | 40.51 | 78.13 | 63.51 | 78.12 | 53.96 | 67.21 | 44.80 | 79.42 | 66.06 | 79.41 |
| **Diff + Dynamic** | **51.61** | **65.46** | **42.12** | **78.58** | **64.42** | **78.57** | **57.94** | **69.88** | **46.77** | **81.54** | **68.49** | **81.54** |
| **Diff + Dynamic + Re-GEM** | **51.61** | **65.46** | **42.12** | **78.58** | **64.42** | **78.57** | **57.94** | **69.88** | **46.77** | **81.54** | **68.49** | **81.54** |
| **Vanilla T5-large** | 50.53 | 64.54 | 40.51 | 78.13 | 63.51 | 78.12 | 53.96 | 67.21 | 44.80 | 79.42 | 66.06 | 79.41 |
| **Diff + Dynamic** | **51.61** | **65.46** | **42.12** | **78.58** | **64.42** | **78.57** | **57.94** | **69.88** | **46.77** | **81.54** | **68.49** | **81.54** |
| **Diff + Dynamic + Re-GEM** | **51.61** | **65.46** | **42.12** | **78.58** | **64.42** | **78.57** | **57.94** | **69.88** | **46.77** | **81.54** | **68.49** | **81.54** |
| **Vanilla T5-base** | 50.53 | 64.54 | 40.51 | 78.13 | 63.51 | 78.12 | 53.96 | 67.21 | 44.80 | 79.42 | 66.06 | 79.41 |
| **Diff + Dynamic** | **51.61** | **65.46** | **42.12** | **78.58** | **64.42** | **78.57** | **57.94** | **69.88** | **46.77** | **81.54** | **68.49** | **81.54** |
| **Diff + Dynamic + Re-GEM** | **51.61** | **65.46** | **42.12** | **78.58** | **64.42** | **78.57** | **57.94** | **69.88** | **46.77** | **81.54** | **68.49** | **81.54** |

Table 7: Results based on different backbone model on MERMAID dataset.

## D.4 Average Runtime

The whole training process for one epoch on two GPUs took approximately 40 minutes including 10 minutes for evaluating difficulties and 30 for fine-tuning.

## E Case Study

In Table 9 and 10, we provide more generated examples from idiomatic expression generation and metaphor generation. Examples from different difficulty levels are selected for comparison. For both tasks, we could observe that most of the baseline models could correctly generate the target idiomatic expressions and metaphors when this example is regarded as easy for the model. However, when the example is selected from examples of medium difficulty levels, some baseline models start to generate wrong idiomatic expressions and metaphors. When it comes to the example from hard difficulty levels, all the baseline models cannot generate the correct idiomatic expressions and metaphors. Only our proposed method could still correctly generate the target idiomatic expressions and metaphors.

Therefore, these generated examples confirm that different examples have different difficulty levels for the models, which justify the need for curriculum learning. Besides, these examples also demonstrate that our proposed method could effectively work by learning from an easy-to-hard order.

| Methods | 1 Epoch | | | | | 5 Epochs | | | | |
|---|---|---|---|---|---|---|---|---|---|---|
| | Acc | BLEU | phrase-BLEU | Rouge | phrase-Rouge | Acc | BLEU | phrase-BLEU | Rouge | phrase-Rouge |
| Vanilla-BART-base | 18 | 56.65 | 37.74 | 69.94 | 40.69 | 25 | 61.87 | 43.82 | 73.27 | 48.81 |
| Diff + Dynamic + Full Data -BART-base | 18 | 56.53 | 37.24 | 70.03 | 40.58 | 25 | 62.02 | 43.76 | 73.17 | 48.66 |
| Diff + Dynamic + Re-GEM + Full Data -BART-base | 19 | 56.83 | 38.02 | 70.11 | 40.79 | 26 | 61.94 | 44.01 | 73.48 | 48.87 |

Table 8: Results based on full data on MAGPIE dataset.

| | | |
|---|---|---|
| **Easy** | Input Sentence | you would n't get anything from anywhere else because they 've cut it **[MASK]** . |
| | Target Sentence | you would n't get anything from anywhere else because they 've cut it **to the bone** . |
| | Vanilla | you wouldn't get anything from anywhere else because they don't cut it *at the end of the day* . |
| | Competence + SL | you wouldn't get anything from anywhere else because they've cut it *by the back door* . |
| | Competence + WR | you would n't get anything from anywhere else because they 've cut it **to the bone** . |
| | Norm-based | you would n't get anything from anywhere else because they 've cut it **to the bone** . |
| | SGCL + fixed | you wouldn't get anything from anywhere else because they don't cut it *at the end of the day* . |
| | SGCL + dynamic | you wouldn't get anything from anywhere else because they've cut it *by the back door* . |
| | Ours | you would n't get anything from anywhere else because they 've cut it **to the bone** . |
| **Medium** | Input Sentence | these were some of the qualities needed to **[MASK]** as a sportsmen even at the more modest levels . |
| | Target Sentence | these were some of the qualities needed to **make the grade** as a sportsmen even at the more modest levels . |
| | Vanilla | these were some of the qualities needed to *rise to the occasion* as a sportsmen even at the more modest levels . |
| | Competence + SL | these were some of the qualities needed to *stand up and be counted* as a sportsmen even at the more modest levels . |
| | Competence + WR | these were some of the qualities needed to *make the killing* as a sportsmen even at the more modest levels . |
| | Norm-based | these were some of the qualities needed to *rise to the occasion* as a sportsmen even at the more modest levels . |
| | SGCL + fixed | these were some of the qualities needed to *make the killing* as a sportsmen even at the more modest levels . |
| | SGCL + dynamic | these were some of the qualities needed to *rise to the occasion* as a sportsmen even at the more modest levels . |
| | Ours | these were some of the qualities needed to **make the grade** as a sportsmen even at the more modest levels . |
| **Hard** | Input Sentence | Are the Americans going **[MASK]** again , or is this an indictment which we should place on trial ? |
| | Target Sentence | Are the Americans going **over the top** again , or is this an indictment which we should place on trial ? |
| | Vanilla | Are the Americans going *behind the scenes* again , or is this an indictment which we should place on trial ? |
| | Competence + SL | Are the Americans going *behind the scenes* again , or is this an indictment which we should place on trial ? |
| | Competence + WR | Are the Americans going *behind the scenes* again , or is this an indictment which we should place on trial ? |
| | Norm-based | Are the Americans going *behind the scenes* again , or is this an indictment which we should place on trial ? |
| | SGCL + fixed | Are the Americans going *through the motions* again , or is this an indictment which we should place on trial ? |
| | SGCL + dynamic | Are the Americans going *behind our backs* again , or is this an indictment which we should place on trial ? |
| | Ours | Are the Americans going **over the top** again , or is this an indictment which we should place on trial ? |

Table 9: A sample of the generated sentences on MAGPIE highlighting the **correct idioms**, and the *wrong idioms*. **Easy** represents the easy example randomly selected from the examples in the start after ranking based on difficulty levels. **Medium** represents the example randomly selected from the examples in the middle after ranking based on difficulty levels. **Hard** represents the example randomly selected from the examples in the final after ranking based on difficulty levels.

| | | |
|---|---|---|
| **Easy** | **Input Sentence** | Whose wrath <V> loomed <V> heavy o'er the trojan towers : |
| | **Target Sentence** | Whose wrath **hung** heavy o'er the trojan towers : |
| | **Vanilla** | Whose wrath **hangs** heavy o'er the trojan towers : |
| | **Competence + SL** | Whose wrath *rises* heavy o'er the trojan towers : |
| | **Competence + WR** | Whose wrath **hangs** heavy o'er the trojan towers : |
| | **Norm-based** | Whose wrath **hangs** heavy o'er the trojan towers : |
| | **SGCL + fixed** | Whose wrath **hangs** heavy o'er the trojan towers : |
| | **SGCL + dynamic** | Whose wrath **hangs** heavy o'er the trojan towers : |
| | **Ours** | Whose wrath **hung** heavy o'er the trojan towers : |
| **Medium** | **Input Sentence** | To that which never can us comfort <V> say <V> . |
| | **Target Sentence** | To that which never can us comfort **bring** . |
| | **Vanilla** | To that which never can us comfort *give* . |
| | **Competence + SL** | To that which never can us comfort **bring** . |
| | **Competence + WR** | To that which never can us comfort *find* . |
| | **Norm-based** | To that which never can us comfort *find* . |
| | **SGCL + fixed** | To that which never can us comfort *make* . |
| | **SGCL + dynamic** | To that which never can us comfort *give* . |
| | **Ours** | To that which never can us comfort **bring** . |
| **Hard** | **Input Sentence** | He hardly knew when waves he <V> used <V> on |
| | **Target Sentence** | He hardly knew when waves he **tossed** on |
| | **Vanilla** | He hardly knew when waves he *plowed* on |
| | **Competence + SL** | He hardly knew when waves he *went* on |
| | **Competence + WR** | He hardly knew when waves he *put* on |
| | **Norm-based** | He hardly knew when waves he *put* on |
| | **SGCL + fixed** | He hardly knew when waves he *rolled* on |
| | **SGCL + dynamic** | He hardly knew when waves he *put* on |
| | **Ours** | He hardly knew when waves he **tossed** on |

Table 10: A sample of the generated sentences on MERMAID highlighting the **correct metaphor**, and the ***wrong metaphor***. **Easy** represents the easy example randomly selected from the examples in the start after ranking based on difficulty levels. **Medium** represents the example randomly selected from the examples in the middle after ranking based on difficulty levels. **Hard** represents the example randomly selected from the examples in the final after ranking based on difficulty levels.