# OpenReview forum: "Non-compositional Expression Generation Based on Curriculum Learning and Continual Learning"
_EMNLP/2023/Conference — EMNLP 2023 Findings_

### Official Review · Reviewer_ZgeL · 2023-07-29

**Soundness:** 3

**Excitement:**

3: Ambivalent: It has merits (e.g., it reports state-of-the-art results, the idea is nice), but there are key weaknesses (e.g., it describes incremental work), and it can significantly benefit from another round of revision. However, I won't object to accepting it if my co-reviewers champion it.

**Paper Topic And Main Contributions:**

This paper focuses on non-compositional expression generation. The challenges of the task lie in the non-compositionality and limited data scale. To address the issues, the authors propose a curriculum-learning-based framework. During training, to mitigate catastrophic forgetting, the authors utilize continual learning. Experiments on both idiomatic expression generation and metaphor generation tasks show the effectiveness of the proposed approach.

**Questions For The Authors:**

See weaknesses

**Reasons To Accept:**

Strengths:
1) The first study on non-compositional expression generation, which may interest the research community.
2) Presenting a framework for this task
3) Better performance

**Reasons To Reject:**

Weaknesses:
1) I learn nothing after reading the method proposed in this work. The curriculum learning and continual learning are both existing methods. A task-specific method may be better.
2) For catastrophic forgetting in curriculum learning, why not train the model with the full data after the curriculum learning stage and how is its performance? In my opinion, this issue may be not serious and experiments also show that it has only a little impact on performance.
3) In line 8, you claimed a large pre-trained language model. However, the BART model you used in experiments is not a large version. So, is there the same trend under the larger pre-trained model or another different architecture (e.g., T5-large)?
4) Lack of many experimental details, e.g., training details, lr, etc

**Reproducibility:**

3: Could reproduce the results with some difficulty. The settings of parameters are underspecified or subjectively determined; the training/evaluation data are not widely available.

**Reviewer Confidence:**

4: Quite sure. I tried to check the important points carefully. It's unlikely, though conceivable, that I missed something that should affect my ratings.

---

> ### Author Rebuttal · Authors · 2023-08-28
>
> 1. While curriculum learning and continual learning are indeed established learning paradigms akin to supervised learning as a learning paradigm, our approach uniquely adapts these paradigms to our specific tasks. Specifically, (1) we introduce new difficulty metrics (Section 3.1) tailored to our challenges and pioneer a novel scheduling strategy for training examples (Section 3.2). Notably, we are the first to highlight the overlooked issue of forgetting in curriculum learning, a significant contribution to the field. Furthermore, rather than merely adopting existing continual learning methods, we have meticulously crafted our method (line383-line412) to cater to our task's nuances, making our method novel and inherently task-specific.
> 2. We respond to this in two stages: (1) Using the full dataset post-curriculum learning would essentially revert to traditional, non-curriculum-based training. While this may mitigate the forgetting problem, it would also eliminate the improvements afforded by  curriculum learning. The crux of curriculum learning lies in reordering training examples from simpler to more complex tasks, contrasting with the random sequencing in standard training. Introducing the full dataset post-curriculum phase would disrupt this deliberate sequencing. (2) Our ablation study (Table 4) underscores the efficacy of our approach: models employing continual learning significantly outperform those without, demonstrated by automated metrics and via a more nuanced human evaluation. According to our ablation study (Table 4), with continual learning, we achieved improvements of 3 points on Accuracy (statistically significant), 3.64 on phrase-BLEU and 2.35 on phrase-ROUGE. According to our human evaluation (Table 5) which is more accurate, we achieved improvements of 1.12 on Meaning, 2.01 on Fitness and 1.11 on Overall.
> 3. Previous research, as cited in [1], has investigated the idiom generation task using various models, including BART and T5 (T5-base). BART-base emerged as the superior performer, leading us to adopt it as our foundational model. It's also essential to recognize that the catastrophic forgetting issue is fundamentally rooted in overfitting. Models tend to lose previously learned information as they adapt to new training samples, precipitating this forgetting challenge. Continual learning serves as our strategy to counteract this. Upscaling the model would exacerbate overfitting, given the heightened complexity, especially in our low-resource setting where sourcing additional training data is impractical. Consequently, even larger architectures like T5-large are susceptible to this pitfall. We've included additional experiments showcasing the efficacy of our proposed method across diverse backbone models:
> When trained for one epoch:
> | Backbone Model | Methods                 | Acc | BLEU  | phrase-BLEU | Rouge | phrase_Rouge |
> |----------------|-------------------------|-----|-------|-------------|-------|--------------|
> | BART-large     | Vanilla                 | 20  | 58.43 | 40.24       | 71.88 | 42.77        |
> | BART-large     | Diff + Dynamic          | 22  | 60.47 | 42.83       | 74.02 | 45.71        |
> | BART-large     | Diff + Dynamic + Re-GEM | 25  | 61.73 | 44.24       | 75.12 | 46.83        |
> | | | | | | | |
> | BART-base      | Vanilla                 | 18  | 56.65 | 37.74       | 69.94 | 40.69        |
> | BART-base      | Diff + Dynamic          | 20  | 59.06 | 40.08       | 71.31 | 43.42        |
> | BART-base      | Diff + Dynamic + Re-GEM | 23  | 60.43 | 43.72       | 72.31 | 45.77        |
> | | | | | | | |
> | T5-large       | Vanilla                 | 18  | 56.21 | 37.13       | 68.74 | 39.23        |
> | T5-large       | Diff + Dynamic          | 20  | 57.94 | 39.85       | 70.03 | 41.73        |
> | T5-large       | Diff + Dynamic + Re-GEM | 22  | 60.74 | 42.23       | 71.94 | 43.72        |
> | | | | | | | |
> | T5-base        | Vanilla                 | 15  | 53.14 | 34.02       | 65.52 | 36.01        |
> | T5-base        | Diff + Dynamic          | 17  | 54.26 | 36.13       | 66.74 | 37.82        |
> | T5-base        | Diff + Dynamic + Re-GEM | 20  | 57.02 | 39.35       | 67.83 | 39.07        |
>
> When trained for 5 epochs:
> | Backbone Model | Methods                 | Acc | BLEU  | phrase-BLEU | Rouge | phrase_Rouge |
> |----------------|-------------------------|-----|-------|-------------|-------|--------------|
> | BART-large     | Vanilla                 | 27  | 62.73 | 45.02       | 74.98 | 50.33        |
> | BART-large     | Diff + Dynamic          | 28  | 64.55 | 47.83       | 77.02 | 53.11        |
> | BART-large     | Diff + Dynamic + Re-GEM | 29  | 66.02 | 49.13       | 78.94 | 55.03        |
> | | | | | | | |
> | BART-base      | Vanilla                 | 25  | 61.87 | 43.82       | 73.27 | 48.81        |
> | BART-base      | Diff + Dynamic          | 26  | 63.20 | 46.84       | 74.57 | 50.17        |
> | BART-base      | Diff + Dynamic + Re-GEM | 28  | 64.36 | 48.02       | 74.94 | 51.82        |
> | | | | | | | |
> | T5-large       | Vanilla                 | 24  | 60.12 | 42.98       | 72.74 | 48.02        |
> | T5-large       | Diff + Dynamic          | 25  | 61.03 | 45.02       | 74.38 | 50.12        |
> | T5-large       | Diff + Dynamic + Re-GEM | 27  | 62.91 | 47.03       | 76.55 | 51.82        |
> | | | | | | | |
> | T5-base        | Vanilla                 | 20  | 56.98 | 39.76       | 69.31 | 44.88        |
> | T5-base        | Diff + Dynamic          | 21  | 58.83 | 41.77       | 70.69 | 46.72        |
> | T5-base        | Diff + Dynamic + Re-GEM | 23  | 61.44 | 44.06       | 72.87 | 49.36        |
>
> 4. To maintain brevity and coherence in the main text, we've allocated detailed experimental specifics to Appendix B and Appendix D, which we will consolidate for clarity of exposition. For transparency and the broader scientific community's benefit, we're also committed to releasing our code once our work receives approval, as mentioned in the abstract.
>
> [1] Zhou, Jianing, et al. "Idiomatic expression paraphrasing without strong supervision." Proceedings of the AAAI Conference on Artificial Intelligence. Vol. 36. No. 10. 2022.

---

### Official Review · Reviewer_vrK3 · 2023-08-02

**Soundness:** 3

**Excitement:**

3: Ambivalent: It has merits (e.g., it reports state-of-the-art results, the idea is nice), but there are key weaknesses (e.g., it describes incremental work), and it can significantly benefit from another round of revision. However, I won't object to accepting it if my co-reviewers champion it.

**Paper Topic And Main Contributions:**

This paper focuses on the non-compositional expression generation task which aims to generate the masked idiom or metaphor in the input sentences. The authors propose a dynamic curriculum learning framework to sort the task samples from easy to hard. The difficulty of each sample is quantified under a special consideration of semantic gaps between the non-compositional expressions and the representations of their contituent words. To alleviate the catastrophic forgetting issue during curriculum learning, they also propose a continual learning strategy (named RE-GEM) inspired by GEM. Experimental results on two benchmark datasets (MAGPIE and MERMAID), and human evaluation show the effectiveness of the proposed method.

**Reasons To Accept:**

- Non-compositional experssion generation is an interesting task
- The experimental results are decent

**Reasons To Reject:**

- The descriptions about implementation details are severely insufficient, making it difficult to reproduce this work. Specifically, the authors only describe a single sentence in Appendix B.
- It is so confused about the backbone of the proposed model (as well as baselines) utill I read the last sentence in Appendix D.4, i.e., "140M for BART(base)". I think the authors should improve the presentation way.
- The experiments are not enough, and the soundness of this paper does not reach the conference level. The authors only conduct experiments with the backbone of the BART base model. However, the size of backbone has a significant influence on the task performance. When the size rises, some issues (such as catastrophic forgetting) may not be challenging. Therefore, only using one backbone with one size is not enough.

**Reproducibility:**

2: Would be hard pressed to reproduce the results. The contribution depends on data that are simply not available outside the author's institution or consortium; not enough details are provided.

**Reviewer Confidence:**

4: Quite sure. I tried to check the important points carefully. It's unlikely, though conceivable, that I missed something that should affect my ratings.

**Typos Grammar Style And Presentation Improvements:**

- Improve the implementation detail section

---

> ### Author Rebuttal · Authors · 2023-08-28
>
> 1. Thank you for highlighting that point. More details can also be found in Appendix D. For clarity, we will consolidate the details. Additionally, we would like to note that we will be releasing our code upon acceptance as stated in the abstract, facilitating a more straightforward implementation.
> 2. We apologize for any confusion caused. Due to page constraints, certain details were abbreviated. We will enhance the presentation for clarity, taking your feedback into consideration.
> 3. Firstly, prior research, e.g. [1], has delved into the task of idiom generation using various models, including BART and T5 (specifically, T5-base). Based on these findings, BART-base outperformed the other models, which influenced our decision to adopt it as our primary model. Furthermore, the catastrophic forgetting issue can be fundamentally characterized as an overfitting problem. When models intensely adapt to new training samples, they inadvertently lose previously acquired knowledge from older examples. This phenomenon leads to the mentioned forgetting problem. Our adoption of continual learning primarily seeks to address this overfitting concern. Increasing the model's size could exacerbate overfitting by amplifying its complexity. Given the resource constraints of our tasks, leveraging more training data to offset this problem isn't feasible. To further substantiate our approach, we've included additional experiments showcasing the efficacy of our proposed method across diverse backbone models:
> When trained for one epoch:
> | Backbone Model | Methods                 | Acc | BLEU  | phrase-BLEU | Rouge | phrase_Rouge |
> |----------------|-------------------------|-----|-------|-------------|-------|--------------|
> | BART-large     | Vanilla                 | 20  | 58.43 | 40.24       | 71.88 | 42.77        |
> | BART-large     | Diff + Dynamic          | 22  | 60.47 | 42.83       | 74.02 | 45.71        |
> | BART-large     | Diff + Dynamic + Re-GEM | 25  | 61.73 | 44.24       | 75.12 | 46.83        |
> | | | | | | | |
> | BART-base      | Vanilla                 | 18  | 56.65 | 37.74       | 69.94 | 40.69        |
> | BART-base      | Diff + Dynamic          | 20  | 59.06 | 40.08       | 71.31 | 43.42        |
> | BART-base      | Diff + Dynamic + Re-GEM | 23  | 60.43 | 43.72       | 72.31 | 45.77        |
> | | | | | | | |
> | T5-large       | Vanilla                 | 18  | 56.21 | 37.13       | 68.74 | 39.23        |
> | T5-large       | Diff + Dynamic          | 20  | 57.94 | 39.85       | 70.03 | 41.73        |
> | T5-large       | Diff + Dynamic + Re-GEM | 22  | 60.74 | 42.23       | 71.94 | 43.72        |
> | | | | | | | |
> | T5-base        | Vanilla                 | 15  | 53.14 | 34.02       | 65.52 | 36.01        |
> | T5-base        | Diff + Dynamic          | 17  | 54.26 | 36.13       | 66.74 | 37.82        |
> | T5-base        | Diff + Dynamic + Re-GEM | 20  | 57.02 | 39.35       | 67.83 | 39.07        |
>
> When trained for 5 epochs:
> | Backbone Model | Methods                 | Acc | BLEU  | phrase-BLEU | Rouge | phrase_Rouge |
> |----------------|-------------------------|-----|-------|-------------|-------|--------------|
> | BART-large     | Vanilla                 | 27  | 62.73 | 45.02       | 74.98 | 50.33        |
> | BART-large     | Diff + Dynamic          | 28  | 64.55 | 47.83       | 77.02 | 53.11        |
> | BART-large     | Diff + Dynamic + Re-GEM | 29  | 66.02 | 49.13       | 78.94 | 55.03        |
> | | | | | | | |
> | BART-base      | Vanilla                 | 25  | 61.87 | 43.82       | 73.27 | 48.81        |
> | BART-base      | Diff + Dynamic          | 26  | 63.20 | 46.84       | 74.57 | 50.17        |
> | BART-base      | Diff + Dynamic + Re-GEM | 28  | 64.36 | 48.02       | 74.94 | 51.82        |
> | | | | | | | |
> | T5-large       | Vanilla                 | 24  | 60.12 | 42.98       | 72.74 | 48.02        |
> | T5-large       | Diff + Dynamic          | 25  | 61.03 | 45.02       | 74.38 | 50.12        |
> | T5-large       | Diff + Dynamic + Re-GEM | 27  | 62.91 | 47.03       | 76.55 | 51.82        |
> | | | | | | | |
> | T5-base        | Vanilla                 | 20  | 56.98 | 39.76       | 69.31 | 44.88        |
> | T5-base        | Diff + Dynamic          | 21  | 58.83 | 41.77       | 70.69 | 46.72        |
> | T5-base        | Diff + Dynamic + Re-GEM | 23  | 61.44 | 44.06       | 72.87 | 49.36        |
>
>
> [1] Zhou, Jianing, et al. "Idiomatic expression paraphrasing without strong supervision." Proceedings of the AAAI Conference on Artificial Intelligence. Vol. 36. No. 10. 2022.

---

### Official Review · Reviewer_gkQD · 2023-08-03

**Typos Grammar Style And Presentation Improvements:** 1. Subscript error in line 317.
2. Th…
**Soundness:** 3

**Excitement:**

4: Strong: This paper deepens the understanding of some phenomenon or lowers the barriers to an existing research direction.

**Paper Topic And Main Contributions:**

The paper aims to improve non-compositional expression generation that have not been explored much.
The main contribution is to propose a new curriculum learning approach adapted to non-compositional expression generation, and conduct continual learning to alleviate catastrophic forgetting in the curriculum learning.


**Questions For The Authors:**

The continual learning part seems to be used to alleviate the problem brought by CL applied to this task, but the works about CL before do not concern this. Does it mean the catastrophic forgetting problem just appear in non-compositional expression generation rather than other tasks?

**Reasons To Accept:**

1. First study on non-compositional expression generation including both metaphors and idioms.
2. The novelty of the proposed method is adequate, which not only reflects on the new task-adaptive CL method but the new continual learning schema.
3. This work also conducts extensive experiments, including comparison to other existing CL method, etc.


**Reasons To Reject:**

In this paper, the task of non-compositional expression generation acts as mask prediction rather than autoregressive generation, which results in a certain degree of limitation.

**Reproducibility:**

4: Could mostly reproduce the results, but there may be some variation because of sample variance or minor variations in their interpretation of the protocol or method.

**Reviewer Confidence:**

4: Quite sure. I tried to check the important points carefully. It's unlikely, though conceivable, that I missed something that should affect my ratings.

---

> ### Author Rebuttal · Authors · 2023-08-28
>
> 1. Our primary objective is to emphasize the generation of non-compositional expressions rather than entire sentence construction. We want to generate non-compositional expressions to fit the given bidirectional context, which could not be accomplished by autoregressive generation. In this sense, we chose mask prediction as our format of the task.
> 2. To date, there hasn’t been any prior research that has specifically highlighted the 'forgetting problem' in curriculum learning. We take pride in pioneering this observation and actively addressing the issue. Our hypothesis is that this problem is inherent in curriculum learning across various tasks. Given that curriculum learning inherently re-orders training samples from simpler to more complex sequences, there's a consequential shift in the distribution of these examples. This shift, we believe, potentially causes the forgetting phenomenon, irrespective of the specific task at hand. For our case of non-compositional expression generation, we propose a specifically designed continual learning to solve this problem. In this sense, the application of continual learning in curriculum learning for other tasks could be a broader research direction, which is out of the scope of this paper.

---

### Meta-Review · Area_Chair_EER9 · 2023-09-14

**Recommendation:** 3

**Metareview:**

This paper addresses an unexplored topic in non-compositional expression generation. The proposed method has been properly evaluated in comparison with various baselines, demonstrating the effectiveness of the proposed approach. A common complaint about the paper is the lack of enough details for the experimental setup. While releasing the code may help in addressing the reproducibility of the results, the paper itself should also contain enough details for understanding the proposed approach and the evaluation settings. I strongly encourage the authors to address the reviewers' concerns in the next revision.

---

### Decision · Program_Chairs · 2023-10-07

**Decision:**

Accept-Findings

**Comment:**

This paper addresses an unexplored topic in non-compositional expression generation. The proposed method has been properly evaluated in comparison with various baselines, demonstrating the effectiveness of the proposed approach. A common complaint about the paper is the lack of enough details for the experimental setup. While releasing the code may help in addressing the reproducibility of the results, the paper itself should also contain enough details for understanding the proposed approach and the evaluation settings. I strongly encourage the authors to address the reviewers' concerns in the next revision.